# The Eosinophil-to-Lymphocyte Ratio Acts as an Indicator for Improvement of Clinical Signs and Itch by Upadacitinib Treatment in Atopic Dermatitis

**DOI:** 10.3390/jcm12062201

**Published:** 2023-03-12

**Authors:** Teppei Hagino, Hidehisa Saeki, Eita Fujimoto, Naoko Kanda

**Affiliations:** 1Department of Dermatology, Nippon Medical School Chiba Hokusoh Hospital, Inzai 270-1694, Japan; 2Department of Dermatology, Nippon Medical School, Tokyo 113-8602, Japan; 3Fujimoto Dermatology Clinic, Funabashi 274-0063, Japan

**Keywords:** atopic dermatitis, upadacitinib, eosinophil-to-lymphocyte ratio, eczema area and severity index, peak pruritus-numerical rating scale

## Abstract

Atopic dermatitis (AD) is a chronic inflammatory skin disease with severe itch. The eosinophil-to-lymphocyte ratio (ELR), neutrophil-to-lymphocyte ratio (NLR), monocyte-to-lymphocyte ratio (MLR), and platelet-to-lymphocyte ratio (PLR) are reported to reflect itch or the severity of AD. We examined if these parameters may act as indicators for therapeutic effects of the Janus kinase 1 inhibitor upadacitinib for patients with AD in real-world clinical practice. Between August 2021 and September 2023, 65 Japanese patients (aged ≥ 12 years) with moderate to severe AD were treated with 15 mg/day of oral upadacitinib, plus twice daily topical corticosteroids. Before treatment, the baseline ELR, NLR, MLR, and PLR levels positively correlated with the eczema area and severity index (EASI), while the baseline NLR and PLR levels positively correlated with the peak pruritus-numerical rating scale (PP-NRS). After upadacitinib treatment, ELR and NLR remarkably decreased at week 4 and the reduced levels were maintained until week 24, in parallel with EASI and PP-NRS, while MLR and PLR transiently reduced at week 4, but returned to baseline levels after week 12. The percent reduction of ELR significantly correlated with the percent reductions of EASI and PP-NRS at weeks 4, 12, and 24 of upadacitinib treatment. ELR may act as an indicator for the improvement of clinical signs and itch by upadacitinib treatment in AD.

## 1. Introduction

Atopic dermatitis (AD) is a chronic inflammatory skin disease with type 2-skewed abnormal immunity, skin barrier impairment, and severe itch [1,2]. Upadacitinib, an oral Janus kinase (JAK) 1 inhibitor, is approved for the treatment of atopic dermatitis (AD) in Japan. We have previously reported that upadacitinib treatment for 12 weeks was well-tolerated and provided comparable or rather superior therapeutic effects for patients with AD in real-world clinical practice, compared with those in previous clinical trials [3].

The neutrophil-to-lymphocyte ratio (NLR), monocyte-to-lymphocyte ratio (MLR), and platelet-to-lymphocyte ratio (PLR) have been identified as indicators of systemic inflammation [4,5,6,7]. In patients with psoriasis, the values of these parameters correlate with the area and severity index of psoriasis (PASI). In particular, NLR was reduced in parallel with PASI by treatment with tumor necrosis factor (TNF)-α inhibitors, indicating that NLR can act as an indicator of disease severity and treatment responses in patients with psoriasis [8,9]. Further, previous studies reported that the eosinophil-to-lymphocyte ratio (ELR), as well as NLR or PLR, in patients with AD, positively correlated with the scoring atopic dermatitis index, an AD severity score [10,11,12]. Recently, Inokuchi-Sakata et al. have reported that ELR correlates with the degree of itch, while NLR correlates with the degree of inflammation and the area of the affected region in patients with AD [13]. However, the transition of these parameters by treatment for AD has not been precisely examined.

In this study, we examined how the values of ELR, NLR, MLR, and PLR might be altered by treatment with upadacitinib in patients with moderate-to-severe AD in real-world practice. We examined if changes in these parameters can reflect the improvement of clinical signs or itch by upadacitinib treatment, as assessed by the eczema area and severity index (EASI) or the peak pruritus-numerical rating scale (PP-NRS), respectively. We also attempted to detect background factors that predict the responsiveness to treatment with upadacitinib for AD.

## 2. Materials and Methods

### 2.1. Study Design and Data Collection

Between August 2021 and December 2022, 65 Japanese patients (aged ≥ 12 years) with moderate-to-severe AD (EASI > 7) living in Chiba Prefecture were treated with oral 15 mg/day of upadacitinib, plus twice daily topical corticosteroids of moderate-to-strongest classes. This study did not involve the patients treated with upadacitinib alone or those with topical corticosteroids alone. The diagnosis of AD was made clinically based on the Japanese Atopic Dermatitis Guidelines 2021 [14]. The patient’s, age, body mass index (BMI), and duration of AD were recorded (Table 1). Medical records were examined retrospectively. This study was conducted based on the Declaration of Helsinki (2004) and was approved by the Ethics Committee of Nippon Medical School Chiba Hokusoh Hospital. Patients provided written informed consent. The values of EASI, ELR, NLR, MLR, and PLR were calculated at weeks 0, 4, 12, and 24 of treatment. The PP-NRS is a patient’s self-reported item and is designed to measure the greatest itch of the past 24 h based on a scale of 0 to 10, with 0 being “non-perceived itch” and 10 being “worst possible itch” [15].

### 2.2. Statistical Analysis

All statistical analyses were performed using EZR (Saitama Medical Center, Jichi Medical School). The Shapiro–Wilk test was used to assess normality. Results are expressed as mean ± standard deviation for variables with a normal distribution, and as median and interquartile range for variables with a nonparametric distribution. Differences in measurements at weeks 0, 4, 12, and 24 of treatment were assessed using repeated measures of analysis of variance for normally distributed variables and using Friedman’s test for non-parametrically distributed variables. Post hoc analysis was performed using Bonferroni correction. Correlations between variables were tested using Spearman’s correlation coefficient. Differences between the two groups were assessed using Student’s *t*-test for variables with a normal distribution, and Mann–Whitney’s U test for variables with a non-parametric distribution. Statistical significance was set at *p* < 0.05. A linear multivariate regression analysis was performed to determine the predictive factors for the high-percentage reduction of EASI or PP-NRS at weeks 4, 12, or 24 of upadacitinib treatment. The analysis included only the variables with a *p* value < 0.05 in univariate analyses and was adjusted for age and sex. Variables with a variance inflation factor >10 were excluded to avoid multicollinearity.

## 3. Results

### 3.1. Demographics and Baseline Characteristics of the Patients

A total of 65 Japanese patients with AD (46 male and 19 female) were enrolled in the study (Table 1). Baseline (week 0) values of EASI, PP-NRS, ELR, NLR, MLR, and PLR are shown in Table 1. In terms of the overall male/female ratio, the result was a higher percentage of males. In terms of age ratio, the result was a high percentage of those aged ≥ 18 years. At baseline, the EASI was median [interquartile range] 20.4 [16.6–32], and PP-NRS was median [interquartile range] 8 [7,8,9,10].

### 3.2. Correlations between Baseline Values of ELR, NLR, MLR, and PLR versus EASI or PP-NRS

Baseline (week 0) values of ELR, NLR, MLR, and PLR positively correlated with baseline EASI (Table 2). Baseline values of NLR and PLR positively correlated with baseline PP-NRS.

### 3.3. Changes in ELR, NLR, MLR, and PLR after Upadacitinib Treatment

EASI and PP-NRS significantly reduced at week 4, and the reduced levels were maintained at weeks 12 and 24 (Figure 1a,b, Appendix A). The ELR, NLR, MLR, and PLR levels significantly reduced at week 4 compared to the baselines (Figure 1c–f, Appendix A). The reduced levels of ELR and NLR were maintained until week 24 (Figure 1c,d), though the ELR level of week 24 was higher than that of week 4 (Figure 1c). The decreases of MLR and PLR at week 4 (Figure 1e,f) were transient, and their levels returned to baseline levels at weeks 12 and 24.

### 3.4. Correlation between Percent Reductions of Laboratory Parameters versus Those of Clinical Indexes

We then analyzed if percent reductions of ELR, NLR, MLR, and PLR may correlate with those of EASI or PP-NRS (Table 3). The percent reduction of ELR positively correlated with those of both EASI and PP-NRS at weeks 4, 12, and 24. The percent reduction of EASI correlated with that of PLR at week 12 and with that of NLR at week 24. The percent reduction of PP-NRS at week 12 positively correlated with those of MLR and PLR. The results indicate that the percent reductions of ELR correlated with those of EASI and PP-NRS throughout the upadacitinib treatment from week 4 to 24. The results indicate that ELR can act as an indicator for the improvement of clinical signs and itch by upadacitinib treatment for AD.

### 3.5. The Relations of Background Factors with Improvement of EASI by Upadacitinib

We then examined if patients’ background factors, sex, age, BMI, disease duration, baseline clinical indexes, or laboratory parameters might influence the improvement of EASI by upadacitinib treatment at weeks 4, 12, or 24 (Table 4). The percent reduction of EASI at week 4 in female patients was higher than that in male patients, indicating a higher treatment response in females. Age positively correlated with the percent reduction of EASI at week 12, and disease duration positively correlated with the percent reduction of EASI at week 24 (Table 4). Baseline values of EASI, PP-NRS, ELR, NLR, MLR, or PLR did not correlate with percent reductions of EASI.

Linear multivariate regression analysis showed that female sex was associated with a high percent reduction of EASI at week 4 (Table 5); however, it failed to detect the association with disease duration or age. The results indicate that female patients may predict great improvement of clinical signs at week 4 of upadacitinib treatment.

### 3.6. The Relations of Background Factors with Improvement of PP-NRS by Upadacitinib

We then examined if patients’ background factors might influence the improvement of PP-NRS by upadacitinib treatment at weeks 4, 12, or 24 (Table 6). Age positively correlated with the percent reduction of PP-NRS at weeks 4 and 12, and baseline NLR positively correlated with the percent reduction of PP-NRS at week 4. Baseline PP-NRS positively correlated with the percent reduction of PP-NRS at weeks 4 and 24 (Table 6).

Linear multivariate regression analysis showed that older age was associated with a high percent reduction of PP-NRS at weeks 4 and 12 (Table 7). High baseline PP-NRS was associated with a high percent reduction of PP-NRS at weeks 4 and 24. However, the analysis failed to detect the association with baseline NLR. The results indicate that older age or higher baseline PP-NRS may predict the great improvement of itch at weeks 4 and 12 or at weeks 4 and 24, respectively.

## 4. Discussion

Before upadacitinib treatment, baseline ELR, NLR, MLR, and PLR significantly correlated with baseline EASI (Table 2). The results were consistent with those of the previous reports [10,11,13], indicating that ELR, NLR, MLR, and PLR may reflect the long-term control status of AD.

Baseline PP-NRS was significantly correlated with baseline NLR and PLR, but not with baseline ELR. This may be related to the contribution of other cells, such as basophils or mast cells, to the baseline pruritus in AD patients, in addition to eosinophils; these cells are a close proximity of sensory nerves in AD lesions and produce substance P and interleukin (IL)-4/13/31, generating itch [16].

The percent reduction of ELR correlated with those of EASI and PP-NRS at weeks 4, 12, and 24. This suggests that the reduction of ELR may reflect the improvement of clinical signs and itch by upadacitinib. The results also indicate that eosinophils may play key roles in the rash and itch of AD and may be the target of upadacitinib treatment. Eosinophils in AD skin lesions secrete granules, such as major basic protein (MBP), eosinophil peroxidase (EPO), and eosinophil cationic protein (ECP) [17]. These granules can induce the damage of keratinocytes, promoting the release of alarmins, such as IL-33 which stimulate basophils, mast cells, and eosinophils. Eosinophil-derived EPO, MBP, ECP, and substance P act on mast cells and induce their degranulation and release of histamine and cytokines, such as TNF-α or type 2 cytokines [18,19]. Eosinophils release a larger amount of IL-31 compared to CD4+ or CD8+T cells, and the released IL-31 further induces chemotaxis and production of IL-31 in eosinophils [20]. The oral JAK1 inhibitor upadacitinib can suppress the effects of IL-31 transducing JAK1/signal transduction and activator of transcription (STAT) signals. This medicine may block the autoactivation of eosinophils, possibly via IL-31 in AD skin lesions.

In the upper dermis of AD lesions, eosinophils are in close proximity to substance P+ nerve fibers and are abundantly producing brain-derived neurotrophic factor (BDNF). The eosinophil-derived BDNF promotes the growth of sensory nerves and induces chemotaxis, and prolongs the survival of eosinophils in an autocrine manner [21]. Conversely, neurons secrete substance P, which induces degranulation and chemotaxis of eosinophils, and suppresses their apoptosis [22]. The activated eosinophils secrete IL-4/IL-13 or IL-31, which bind IL-4Rα or IL-31RA, respectively, on sensory nerve endings and stimulate the JAK1/STAT pathway, generating an itch sensation [23,24]. The eosinophil-derived IL-31 also promotes the growth of sensory nerves [25]. The oral JAK1 inhibitor upadacitinib might suppress the effects of IL-31 or IL-4/IL-13 and block the communication between nerves and eosinophils, leading to the suppression of pruritus.

The change of NLR, MLR, and PLR did not correlate with that of EASI or PP-NRS in upadacitinib treatment, except for the transient correlation (Table 3). Granulocyte colony-stimulating factor (G-CSF), granulocyte-macrophage colony-stimulating factor (GM-CSF), or thrombopoietin promotes the granulopoiesis, granulopoiesis/monocytopoiesis [26], or thrombopoiesis, respectively, in bone marrow [27]. Though these cytokines, G-CSF, GM-CSF, and thrombopoietin transduce signals through JAK2, and the JAK1 inhibitor upadacitinib can suppress JAK2 at tissue concentrations generated by the standard intake [28]. Therefore, upadacitinib may suppress the effects of these cytokines and, resultantly, reduce the numbers of peripheral neutrophils, monocytes, and platelets within the range, allowing the continuation of treatment. However, the decrease of NLR, MLR, and PLR appears independent of the improvement of clinical signs or itch, indicating that neutrophils, monocytes, or platelets may not directly mediate the upadacitinib-induced improvement of rash or itch. We should further examine if this trend can be seen in other treatments for AD.

Older age was associated with a high percent reduction of PP-NRS at weeks 4 and 12, indicating that older age may predict better improvement of itch by upadacitinib. This may possibly be because older patients might show reduced type 2 activities compared to younger patients [29]; in adult patients with AD, the levels of IL-31 or IL-13 (cytokines generating itch) in the skin lesions are reduced with ageing [29,30]. Thus, older patients with AD might be more susceptible to the inhibitory effects of upadacitinib on the cytokines related to itch.

High baseline PP-NRS was associated with a high percent reduction of PP-NRS at weeks 4 and 24, indicating that high baseline PP-NRS may predict better improvement of itch by upadacitinib. This is possibly because patients with higher PP-NRS values before treatment may have more room to be improved by upadacitinib compared to the patients with lower PP-NRS. A similar trend can be seen in patients with psoriasis; patients with higher pretreatment PASI result in a higher percent reduction of PASI by treatment with biologics [31].

Female sex was associated with a high percent reduction of EASI at week 4, indicating that female patients may predict better improvement of rash by upadacitinib than male patients. However, the bias caused by the male-preponderance (71% of the patients) and small sample size cannot be denied in this study. The sex difference in the response to upadacitinib treatment should be further examined strictly in larger cohorts with a uniform assignment of sexes.

This study has several limitations. First, this study evaluated the efficacy of only 15 mg/day of upadacitinib. The results for 30 mg/day upadacitinib should be further examined. Second, this study is a retrospective one with a small sample size. Third, this study included patients treated with oral upadacitinib plus topical corticosteroids, but not those with oral upadacitinib alone or with topical corticosteroids alone. Therefore, it is unclear whether upadacitinib plus topical corticosteroids additively or synergistically improved clinical signs and itch. Further prospective study with a larger cohort should compare the therapeutic effects of upadacitinib alone, topical corticosteroids alone, and both treatments.

## 5. Conclusions

Before treatment, ELR, NLR, MLR, and PLR positively correlated with EASI, while NLR and PLR positively correlated with PP-NRS. After upadacitinib treatment, ELR and NLR remarkably decreased throughout the treatment period from weeks 4 to 24 in parallel with EASI and PP-NRS, while the decrease of MLR and PLR at week 4 was transient. The percent reduction of ELR significantly correlated with those of both EASI and PP-NRS at weeks 4, 12, and 24 of upadacitinib treatment. Linear multivariate regression analyses revealed that female sex was associated with a high percent reduction of EASI at week 4; older age was associated with a high percent reduction of PP-NRS at weeks 4 and 12; and high baseline PP-NRS was associated with a high percent reduction of PP-NRS at weeks 4 and 24. ELR may act as an indicator for the improvement of clinical signs and itch by upadacitinib treatment for AD.

## Figures and Tables

**Figure 1 jcm-12-02201-f001:**
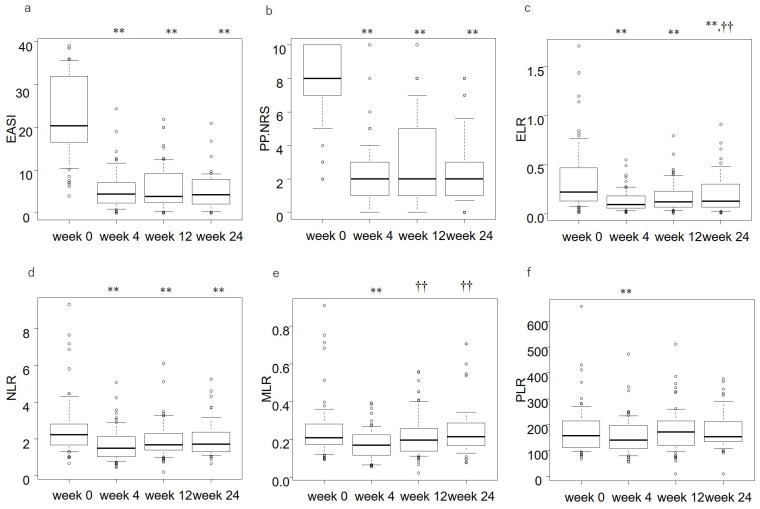
The transition of the eczema area and severity index (EASI) (**a**), peak pruritus-numerical rating scale (PP-NRS) (**b**), eosinophil-to-lymphocyte ratio (ELR) (**c**), neutrophil-to-lymphocyte ratio (NLR) (**d**), monocyte-to-lymphocyte ratio (MLR) (**e**), and platelet-to-lymphocyte ratio (PLR) (**f**) at weeks 0, 4, 12, or 24 of upadacitinib treatment in patients with atopic dermatitis (*n* = 65). The thick lines in the boxes are medians, and the boxes are interquartile ranges. ** *p* < 0.01 versus the value of week 0; †† *p* < 0.01 versus the value of week 4, analyzed by Friedman’s test.

**Table 1 jcm-12-02201-t001:** Demographics and baseline characteristics of patients with atopic dermatitis.

Sex, *n* (%)	
Male	46 (71)
Female	19 (29)
Age (years) ^a^	34 [15–50]
Age group, *n* (%)	
<18 years	21 (32)
≥18 years	44 (68)
Body mass index (kg/m^2^) ^b^	23.74 ± 4.1
Disease duration (years) ^a^	20 [13–40]
Clinical indexes	
EASI ^a^	20.4 [16.6–32]
PP-NRS ^a^	8 [7–10]
Laboratory parameters	
ELR ^a^	0.22 [0.13–0.47]
NLR ^a^	2.23 [1.65–2.81]
MLR ^a^	0.21 [0.17–0.28]
PLR ^a^	156.6 [112.4–214]

^a^ Data provided as the median [interquartile range]. ^b^ Data provided as the mean ± standard deviation. EASI, eczema area and severity index; PP-NRS, peak pruritus-numerical rating scale; ELR, eosinophil-to-lymphocyte ratio; NLR, neutrophil-to-lymphocyte ratio; MLR, monocyte-to-lymphocyte ratio; PLR, plate-let-to-lymphocyte ratio.

**Table 2 jcm-12-02201-t002:** Correlation between baseline values of laboratory parameters versus those of clinical indexes in patients with atopic dermatitis (*n* = 65).

	ELR		NLR		MLR		PLR	
	Rho	*p*	Rho	*p*	Rho	*p*	Rho	*p*
EASI	0.448	0.000233 **	0.297	0.0163 *	0.391	0.00129 **	0.403	0.000878 **
PP-NRS	0.209	0.1	0.347	0.00463 **	0.211	0.0913	0.309	0.0124 *

Correlations between variables were examined using Spearman’s correlation coefficient. * Statistically significant at *p* < 0.05, ** *p* < 0.01. ELR, eosinophil-to-lymphocyte ratio; NLR, neutrophil-to-lymphocyte ratio; MLR, monocyte-to-lymphocyte ratio; PLR, platelet-to-lymphocyte ratio; EASI, eczema area and severity index; PP-NRS, peak pruritus-numerical rating scale.

**Table 3 jcm-12-02201-t003:** Correlations between percent reductions of laboratory parameters versus those of clinical indexes at weeks 4, 12, or 24 of treatment with upadacitinib plus topical corticosteroids in patients with atopic dermatitis (*n* = 65).

		Week 4		Week 12		Week 24	
Clinical Indexes	Laboratory Parameters	Rho	*p*	Rho	*p*	Rho	*p*
EASI	ELR	0.476	0.0000921 **	0.465	0.00016 **	0.403	0.00732 **
	NLR	−0.0333	0.794	0.191	0.133	0.396	0.00586 **
	MLR	0.0462	0.717	0.197	0.122	0.173	0.245
	PLR	0.0747	0.557	0.291	0.0206 *	0.206	0.165
PP-NRS	ELR	0.402	0.00119 **	0.43	0.00054 **	0.43	0.0045 **
	NLR	0.00119	0.0992	0.182	0.154	0.2	0.188
	MLR	0.182	0.151	0.249	0.0494 *	0.106	0.487
	PLR	0.103	0.418	0.255	0.0433 *	0.135	0.376

Correlations between variables were examined using Spearman’s correlation coefficient. * Statistically significant at *p* < 0.05, ** at *p* < 0.01. EASI, eczema area and severity index; PP-NRS, peak pruritus-numerical rating scale; ELR, eosinophil-to-lymphocyte ratio; NLR, neutrophil-to-lymphocyte ratio; MLR, monocyte-to-lymphocyte ratio; PLR, platelet-to-lymphocyte ratio.

**Table 4 jcm-12-02201-t004:** The relations of patients’ background factors with the percent reduction of EASI at weeks 4, 12, or 24 of upadacitinib treatment for patients with atopic dermatitis (*n* = 65).

Background Factors	Percent Reduction of EASI at Week 4			Percent Reduction of EASI at Week 12			Percent Reduction of EASI at Week 24		
Sex (*n*)	Male (46)	Female (19)	*p*	Male (46)	Female (19)	*p*	Male (46)	Female (19)	*p*
	74.14 [68.47–79.79] ^a^	83.8 [74.69–93.49] ^a^	0.0227 *	70.99 [63.74–86.25] ^a^	87.65 [73.75–93.08] ^a^	0.119	81.09 [69.27–91.76] ^a^	82.38 [75.16–91.72] ^a^	0.634
	Rho	*p*		Rho	*p*		Rho	*p*	
Age	0.159	0.206		0.246	0.0496 *		0.26	0.0687	
BMI	0.0256	0.842		0.0359	0.782		−0.0413	0.776	
Disease duration	0.066	0.61		0.176	0.165		0.295	0.0375 *	
ELR	0.15	0.241		0.0279	0.829		−0.01	0.946	
NLR	−0.156	0.215		−0.0136	0.915		0.0652	0.653	
MLR	−0.109	0.389		−0.112	0.378		−0.166	0.249	
PLR	−0.12	0.341		−0.0055	0.966		−0.0297	0.838	
EASI	−0.0791	0.531		−0.0718	0.573		−0.0327	0.822	
PP-NRS	−0.0351	0.781		−0.0292	0.819		−0.038	0.793	

^a^ Data provided as the median [interquartile range], assessed by Mann–Whitney U test. Correlations between variables were examined using Spearman’s correlation coefficient. * Statistically significant at *p* < 0.05. EASI, eczema area and severity index; PP-NRS, peak pruritus-numerical rating scale; ELR, eosinophil-to-lymphocyte ratio; NLR, neutrophil-to-lymphocyte ratio; MLR, monocyte-to-lymphocyte ratio; PLR, platelet-to-lymphocyte ratio.

**Table 5 jcm-12-02201-t005:** The predictive factors for the percent reduction of eczema area and severity index (EASI) at weeks 4, 12, or 24 of upadacitinib treatment assessed by linear multivariate regression analysis in patients with atopic dermatitis (*n* = 65).

	Percent Reduction of EASI at Week 4				Percent Reduction of EASI at Week 12				Percent Reduction of EASI at Week 24			
	β Coefficient	Standard Error	*t*	*p*	β Coefficient	Standard Error	*t*	*p*	β Coefficient	Standard Error	*t*	*p*
Intercept	78.0768	4.3676	17.876	<2 × 10^−16^	67.89	6.8924	9.85	3.17 × 10^−14^	63.6237	13.6684	4.655	2.77 × 10^−5^
Age	0.129155	0.0838	1.541	0.1284	0.2282	0.1357	1.682	0.0977	0.19	0.4176	0.455	0.651
Sex [T.M]	−9.8267	3.845	−2.556	0.0131 *	−3.452	5.9502	−0.58	0.564	−5.4743	11.3297	−0.483	0.631
Disease duration	NA								0.2251	0.5462	0.412	0.682

* Statistically significant at *p* < 0.05. NA, not applicable.

**Table 6 jcm-12-02201-t006:** The relations of patients’ background factors with the percent reduction of the peak pruritus-numerical rating scale (PP-NRS) at weeks 4, 12, or 24 of upadacitinib treatment for patients with atopic dermatitis (*n* = 65).

Background Factors	Percent Reduction of PP-NRS at Week 4			Percent Reduction of PP-NRS at Week 12			Percent Reduction of PP-NRS at Week 24		
Sex (*n*)	Male (46)	Female (19)	*p*	Male (46)	Female (19)	*p*	Male (46)	Female (19)	*p*
	70.71 [57.14–87.5] ^a^	66.67 [50–82.5] ^a^	0.675	60 [33.33–85.71] ^a^	75 [50–81.67] ^a^	0.517	60 [33.33–85.71] ^a^	70 [44.44–83.33] ^a^	0.852
	Rho		*p*	Rho		*p*	Rho		*p*
Age	0.423		0.00045 **	0.329		0.00803 **	0.195		0.184
BMI	0.0958		0.455	−0.0173		0.894	−0.113		0.443
Disease duration	0.143		0.256	0.16		0.205	0.0452		0.76
ELR	0.0667		0.603	0.0229		0.86	0.124		0.406
NLR	0.27		0.0294 *	0.104		0.415	−0.00103		0.994
MLR	0.169		0.177	0.0507		0.691	−0.0471		0.751
PLR	0.128		0.311	0.135		0.287	5.44 × 10^−05^		1
EASI	0.0795		0.529	−0.0285		0.823	0.0757		0.609
PP-NRS	0.402		0.000915 **	0.237		0.0595	0.422		0.00283 **

^a^ Data provided as the median [interquartile range], assessed by Mann-Whitney U test. Correlations between variables were examined using Spearman’s correlation coefficient. * Statistically significant at *p* < 0.05. ** Statistically significant at *p* < 0.01. BMI, body mass index; ELR, eosinophil-to-lymphocyte ratio; NLR, neutrophil-to-lymphocyte ratio; MLR, monocyte-to-lymphocyte ratio; PLR, platelet-to-lymphocyte ratio; EASI, eczema area and severity index.

**Table 7 jcm-12-02201-t007:** The predictive factors for the percent reduction of the peak pruritus-numerical rating scale (PP-NRS) at weeks 4, 12, or 24 of upadacitinib treatment assessed by linear multivariate regression analysis in patients with atopic dermatitis (*n* = 65).

	Percent Reduction of PP-NRS at Week 4				Percent Reduction of PP-NRS at Week 12				Percent Reduction of PP-NRS at Week 24			
	β Coefficient	Standard Error	*t*	*p*	β Coefficient	Standard Error	*t*	*p*	β Coefficient	Standard Error	*t*	*p*
Intercept	27.978	11.06	2.53	0.0141	37.1177	11.0973	3.345	0.00141	−15.82423	20.90884	−0.757	0.45319
Age	0.3048	0.1455	2.095	0.0404 *	0.5733	0.2184	2.625	0.01095 *	0.02703	0.2634	0.103	0.91873
Sex [T.M]	−3.0387	6.1739	−0.492	0.6244	−1.2801	9.5803	−0.134	0.89415	−8.85103	11.96131	−0.74	0.46325
Baseline NLR	2.2744	1.885	1.207	0.2323	NA							
Baseline PP-NRS	3.1617	1.3547	2.334	0.023 *	NA				10.54516	2.49854	4.221	0.00012 **

* Statistically significant at *p* < 0.05. ** Statistically significant at *p* < 0.01. NA, not applicable; NLR, neutrophil-to-lymphocyte ratio.

## Data Availability

Not applicable.

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
