# Peer review of "The Eosinophil-to-Lymphocyte Ratio Acts as an Indicator for Improvement of Clinical Signs and Itch by Upadacitinib Treatment in Atopic Dermatitis"

_jcm, 2023, doi:10.3390/jcm12062201_

Round 1
Reviewer 1 Report
The article is properly structured and well-developed. It highlights a topic of the recent debate on the use of new jak1 inhibitor drugs. You may consider making the abstract more concise to make the take-home message more impactful. The study has limitations to be found in the smallness of the sample size examined, however these limitations are sufficiently explained. You could consider making the discussion less repetitive, especially when talking about the physiological mechanisms of the immune system cells related to the pathologies considered. A general revision of sentences in English is recommended, replacing sentences that are too long with shorter sentences separated by appropriate punctuation. It is advisable to replace where the term “analyzed” occurs with “analyzed”.
lines33-35 you need some references, such as: doi: 10.1111/exd.14276. and doi: 10.1111/jdv.17964.
In conclusion, the paper will be publishable after these minor revisions.
Author Response
添付資料をご覧ください。

Reviewer 2 Report
Atopic dermatitis (AD) is a chronic inflammatory skin disease with genetic predisposition and severe itching. There are many factors that induce atopic dermatitis, including climate, living environment, abnormal drug response, mental factors, etc. For dermatitis with mild symptoms and small area, drugs containing corticosteroids can be selected. However, for repeated attacks and large areas of dermatitis, if drugs containing corticosteroids are used frequently or in large quantities, there will be systemic and local side effects. Therefore, it is of great significance to choose non-corticosteroid drugs to treat specific dermatitis. Upadacitinib is an oral Janus kinase (JAK) 1 selective inhibitor used to treat moderate and severe rheumatoid arthritis, including psoriasis, specific dermatitis and other patients with poor response to treatment. In this paper, 65 Japanese patients with moderate and severe AD (aged ≥ 12 years) were observed to receive oral upadacitinib 15 mg/day plus topical corticosteroids twice a day. The results showed that ELR and NLR were significantly decreased at the 4th week after treatment with Upadacetinib, and the level of simultaneous reduction with EASI and PP-NRS remained at the 24th week. However, MLR and PLR temporarily decreased in the 4th week and returned to the baseline level after the 12th week. At the 4th, 12th and 24th weeks, the percentage of decrease in ELR was significantly correlated with the percentage of decrease in EASI and PP-NRS. Linear multiple regression analysis showed that women were associated with a high percentage reduction in EASI at week 4; At the 4th and 12th weeks, the older the age, the higher the reduction rate of PP-NRS.
There are some important questions in the manuscript, and hope the author can answer them.
1) In order to make the data more intuitive, it is recommended that the author use a histogram to display the data of each group, especially Figure 1.
2) In the author’s study design, group comparison was not rigorous. This study set upadacitinib plus local corticosteroid treatment group twice a day, but did not use upadacitinib treatment group and corticosterone treatment group alone. Therefore, it is unclear whether upadacitinib or corticosterone contributed to the clinical observation effect, or whether there is a synergistic effect between the two?
3) There is not a picture related to atopic dermatitis in the data of this study. The ideal situation is to display the photos before and after drug treatment at the same time. If there is a pathological section of the biopsy tissue, it will be more convincing.
4) The levels of ELR, NLR, MLR, and PLR are positively correlated with EASI, but this study does not mention how to count eosinophils, neutrophils, monocytes, and lymphocytes, so it is difficult to tell the changes of these cells.
